# Probing Layered Tissues by Backscattering Mueller Matrix Imaging and Tissue Optical Clearing

**Qizhi Lai** [1,†], **Tongjun Bu** [1,†], **Tongyu Huang** [2], **Yanan Sun** [3], **Yi Wang** [3,*] **and Hui Ma** [1,2,*]

1   Shenzhen International Graduate School, Tsinghua University, Shenzhen 518055, China; lqz21@mails.tsinghua.edu.cn (Q.L.); btj20@mails.tsinghua.edu.cn (T.B.)

2   Department of Biomedical Engineering, Tsinghua University, Beijing 100084, China; hty19@mails.tsinghua.edu.cn

3   Experimental Research Center, China Academy of Chinese Medicine Science, Beijing 100091, China; sunyanan@merc.ac.cn

\*   Correspondence: wangyi@merc.ac.cn (Y.W.); mahui@tsinghua.edu.cn (H.M.)

†   These authors contributed equally to this work.

**Abstract:** Polarization imaging is a label-free and non-invasive technique that is sensitive to microstructure and suitable for probing the microstructure of living tissues. However, obtaining deep-layer information from tissues has been a challenge for optical techniques. In this work, we used tissue optical clearing (TOC) to increase optical penetration depth and characterize the layered structures of tissue samples. Different tissue phantoms were constructed to examine changes in the polarization features of the layered structure during the TOC process. We found that depolarization and anisotropy parameters were able to distinguish between single-layer and double-layer phantoms, reflecting microstructural information from each layer. We observed changes in polarization parameter images during the TOC process and, by analyzing different regions of the images, explained the sensitivity of these parameters to double-layer structures and analyzed the influence of oblique incident illumination. Finally, we conducted TOC experiments on living skin samples, leveraging the experience gained from phantom experiments to identify the double-layer structure of the skin and extract features related to layered structures. The results show that the combination of backscattering polarization imaging and tissue optical clearing provides a powerful tool for the characterization of layered samples.

**Keywords:** polarization; backscattering; Mueller matrix; layered structure

## 1. Introduction

Polarization imaging techniques have been demonstrated as a powerful tool for probing the complex structures of biological tissues [1–8]. Backscattering Mueller matrix imaging in particular has shown many advantages as a non-contact, noninvasive, and in situ technique for living samples [9,10].

Layered structures are very common in biological tissues. However, superimposing two layers of even the simplest anisotropic property can lead to very complex Muller matrices [11]. Identifying the layered structures and obtaining information of individual layers will greatly enhance the capability of backscattering polarimetry to characterize complex biological tissues, such as the anisotropic properties, thickness, and the scattering coefficient of each layer.

One of the methods to reach different penetration depth is by multi-wavelength backscattering Muller imaging [12]. By comparing polarization data at different wavelengths using machine learning, one can extract the polarization parameters sensitive to multilayer structures.

Tissue optical clearing (TOC) can also reveal layered information about biological tissues [13–19]. The TOC technique reduces the refractive index difference between the

scatterers and the surrounding medium by applying optical clearing agents (OCAs) at the tissue surface, thus reducing the scattering of light and increasing the penetration depth of light in the tissue [20,21], which paves the way for probing layered information about biological tissues.

In a previous work, we used a fast Mueller matrix imaging system to take backscattering images of a composite porcine adipose and porcine skin double-layer phantom [22]. Dynamic curves of different polarization parameters of single- and double-layer samples during the TOC process indicated that the combination of fast backscattering Mueller matrix imaging with TOC technology can serve as an effective tool for quantitatively characterizing the structures of layered tissues.

In this study, we developed more sophisticated double-layer phantoms and examined in detail how dynamic behaviors of different polarization parameters during TOC are correlated to the microstructural features of different layers. We obtained parameters sensitive to layered structures and successfully extracted structural information from the upper and lower layers, explained why different polarization parameters can specifically identify double-layer structures, and analyzed the impact of oblique incident illumination on polarization parameters. Finally, we conducted TOC experiments on the skin of live mice for potential applications for the assessment of drug effects.

## 2. Materials and Methods

### 2.1. Experimental Setup

The fast-backscattering polarimetry imaging system is shown in Figure 1a,b. The system is based on a cage design (RayCage Photoelectric Technology Co., Ltd., Zhenjiang, China).

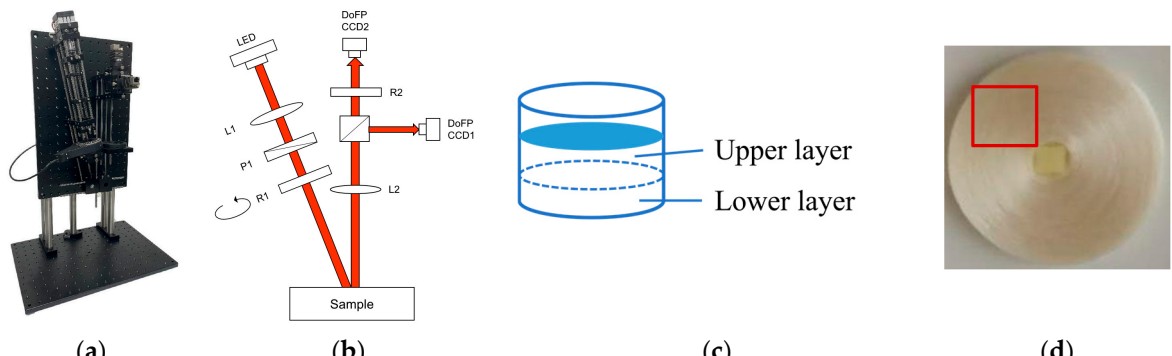

**Figure 1.** Experimental setup and materials. (**a**) Photograph of experimental setup. (**b**) Schematic of experimental setup. (**c**) A schematic diagram of the double-layer phantoms. (**d**) Photograph of concentrically aligned silk fibers.

Emission from the LED (10 W, 633 nm, Cree, Shanghai, China) is converged into parallel light after passing through the lens L1 (Daheng Optic, Beijing, China). Then, the illumination beam passes through the polarization state generator (PSG), which consists of a fixed polarizer P1 (extinction ratio > 1000:1, Daheng Optic, Beijing, China) and a rotatable quarter-wave plate R1 (LBTEK Optic, Changsha, China) placed on a motorized rotation stage (PRM1/Z8, Thorlabs, Newton, NJ, USA). The incident light reaches the sample at a 15° oblique angle, creating a nearly homogeneous illumination area of 5 cm diameter at the sample. The angle between polarization state generator and polarization state analyzer can avoid the influence of reflected light from the sample surface on measurements without causing distortion in the imaging results.

Backscattered light from the sample is collected by the lens L2 (LBTEK Optic, China) along the normal direction of the sample surface to avoid the surface reflection, and then it is received by the dual division of focal plane (DoFP) polarimeters for image detection [23]. The DoFP polarimeters are capable of capturing the complete Stokes vector

in a single shot. With just four measurements, Mueller matrix images of the sample can be obtained. The dual DoFP image detector consists of a non-polarized beam splitter (NPBS, Thorlabs, Newton, NJ, USA) prism, a fixed quarter-wave plate R2 (LBTEK Optic, China) in the transmission channel, and the two paired DoFPs (Lucid Vision Labs, Vancouver, BC, Canada) for both the transmission and reflection channels of the NPBS. Both DoFPs have the same field of view (FOV), exposure time, and resolution (2448 × 2048 pixels for 1.2 × 1.0 cm$^2$ imaging area). The system is calibrated by measuring air as a standard sample. The system takes 15 s to acquire a Mueller matrix image. By measuring air as the standard sample, errors are estimated as less than 1%.

### 2.2. Samples

### 2.2.1. Design of Tissue Phantoms

The use of tissue phantoms offers improved experimental control, minimizing biological variability and environmental influences. Additionally, the standardization of experiments is facilitated by the reproducibility and consistency achievable with tissue phantoms. Based on the characteristics of skin structure, we constructed various tissue phantoms, including single-layer phantoms and double-layer phantoms with distinct features in the upper and lower layers.

We used the isotropic and homogeneous milk as one layer, $S_m$, which was also used as a control group in experiments [24,25]. The thickness and concentration of the milk layer can be adjusted easily. By adding glycerol to milk, we observed the dynamic process of TOC in single-layer samples.

We also constructed double-layer phantoms to simulate the layered structure of the skin, as shown in Figure 1c. The upper layer of the phantoms was the milk, mimicking the isotropic structure of the skin. The lower layers of the phantoms were tissues or other materials submerged in the milk to mimic different types of double-layer phantoms, as shown in Table 1. The thickness and scattering coefficient of the upper layer, and the structural characteristics of the anisotropic medium in the lower layer were easy to control and adjust. Different lower-layers samples corresponded to distinct structural features.

**Table 1.** Sample construction.

| Type | Name | Descriptions |
|------|------|-------------|
| Single-layer sample | $S_m$ | 5 cm milk |
| Double-layer samples | $D_{m-s}$ | Upper layer: 0.3 cm milk<br>Lower layer: concentrically aligned silk fibers |
| | $D_{m-m}$ | Upper layer: 0.3 cm milk<br>Lower layer: porcine skeletal muscle |
| | $D_{m-a}$ | Upper layer: 0.3 cm milk<br>Lower layer: porcine adipose |

The lower layer of $D_{m-m}$ was the anisotropic porcine skeletal muscle. $D_{m-m}$ corresponded to a double-layer structure with isotropic upper layer and anisotropic lower layer. The lower layer of $D_{m-a}$ was isotropic porcine adipose tissue. $D_{m-a}$ and $D_{m-m}$ can be used to evaluate the influence of anisotropy in the lower layer on the results of MMI. The lower layer of $D_{m-s}$ was concentrically arranged silk fibers (shown in Figure 1d), whose anisotropic properties varied with the radius and azimuth angle. The TOC of $D_{m-s}$ further tested the ability of MMI to obtain information from the lower layer and evaluate the reliability of the data. We placed and fixed the samples in special containers separately, then added an equal amount of milk, constituting different double-layer phantoms. Tissue optical clearing was performed on these phantoms to obtain deep-layered polarization information of the samples.

2.2.2. TOC Procedures of Tissue Phantoms

We used glycerol as an optical clearing agent for tissue optical clearing. Optical clearing agents elevate the interstitial refractive index of scattering medium to match the refractive index of scatterers, thereby reducing light scattering in the medium and increasing penetration depths of the photons. Glycerol, with its low cost and good biocompatibility, is a commonly employed clearing agent in experiments. Furthermore, as a mild OCA, glycerol generally does not cause allergic reactions. Adding glycerol increases the interstitial refractive index among the oil droplets scatterers in milk, achieving tissue optical clearing effects.

The milk was placed in an opaque cylindrical container of 8 cm in diameter. The thickness of the milk was 5 cm. To investigate the effect of the TOC process, we substituted a small amount of milk with an equal amount of pure glycerol and stirred the liquid for thorough mixing. The mixed liquid was immediately measured using the backscattering Mueller matrix imaging system. Such a procedure was repeated to obtain 15 sets of data during TOC, corresponding to different densities of the scatterers and different interstitial refractive indices.

For the bilayer samples, we first measured the Mueller matrix of the double-layer samples before the TOC process and then performed the TOC operation on the upper milk layer. We substituted a small amount of milk with an equal amount of pure glycerol and stirred the liquid to mix it well. The samples were measured immediately after the operation. Such a procedure was repeated to obtain 15 sets of data.

The refractive index of milk is 1.35, and the refractive index of glycerol is 1.47. The volume fractions of milk and glycerol in the mixed liquid during TOC were known, and we calculated the refractive index of the milk and glycerol mixture based on the Lorentz–Lorenz equation [26] to quantify the value of interstitial refractive index or the degree of TOC.

2.2.3. Living Skin Samples

The samples for the living skin experiment were six 8-week-old C57BL/6 mice provided by the Medical Experimental Center of the China Academy of Chinese Medical Sciences. Using mice allowed for practical advantages, including ease of experimental control and shorter lifecycles, facilitating efficient data acquisition and experimental design.

Prior to the experiment, the mice were depilated to expose the skin in the imaging area. During the experiment, isoflurane gas anesthesia was used, and the mice were fixed with a holder to prevent movement. The region of interest (ROI) on the back of the mice was marked using a marker pen. For the TOC experiment, 50 μL of glycerol was evenly applied to the ROI of the mice. After a 20n s interval, the residual glycerol on the surface was removed, and the measurement was initiated using the backscattering Mueller matrix imaging system. The use of animals in the experiment was approved by the Animal Ethics Committee of the Experimental Research Center of the China Academy of Chinese Medicine Science.

*2.3. Polarization Feature Extraction Techniques*

The Mueller matrix contains all the polarimetric information of the sample. Apart from Mueller matrix elements, large numbers of polarization basis parameters (PBPs) have been proposed to characterize polarization features [27]. These PBPs are functions of the Mueller matrix elements but tend to have more explicit connections to the optical and microstructural properties of the media. In this article, PBPs from Mueller matrix transformation (MMT) parameters $b$ and $t_1$ were used. The definitions of parameters are shown in Equation (1).

$$
\begin{aligned}
b &= \tfrac{1}{2}(m_{22} + m_{33}) \\
t_1 &= \tfrac{1}{2}\sqrt{(m_{22} - m_{33})^2 + (m_{23} + m_{32})^2}
\end{aligned}
\tag{1}
$$

The Mueller matrix is a $4 \times 4$ matrix, and $m_{ij}$ represents the element in the *i*-th row and *j*-th column of the matrix.

MMT parameter *b* is related to polarization maintaining property or inversely to depolarization, where a smaller value indicates a higher degree of depolarization. The parameter $t_1$ represents the degree of anisotropy, which includes contributions from both the aligned fibrous structure and the birefringence in the sample [27].

## 3. Results and Discussions

### *3.1. TOC Dynamics of Different Layered Phantoms*

Investigating layered phantom provides evidence for interpreting the results of living skin experiments. Therefore, we initially conducted TOC experiments on phantoms and recorded the changes in different samples with TOC using our backscattering Mueller matrix imaging system.

### 3.1.1. Variation of Mueller Matrix during TOC

We took the backscattering MM images during the TOC processes, then took the average of pixels in the ROI and plotted the Mueller matrix curves of different samples, as shown in Figure 2. To avoid crosstalk among different silk orientations, we selected ROIs of the $D_{m\text{-}s}$ sample with similar silk orientation, as shown in Figure 1d, where the red square area represents the ROI. The horizontal axis represents the interstitial refractive index, which measures the extent of TOC. We found that the curves of different samples differed mainly in the diagonal elements. The characteristic temporal trends of a single-layer and a double-layer, or different double-layer samples showed distinctively different responses during the TOC process.

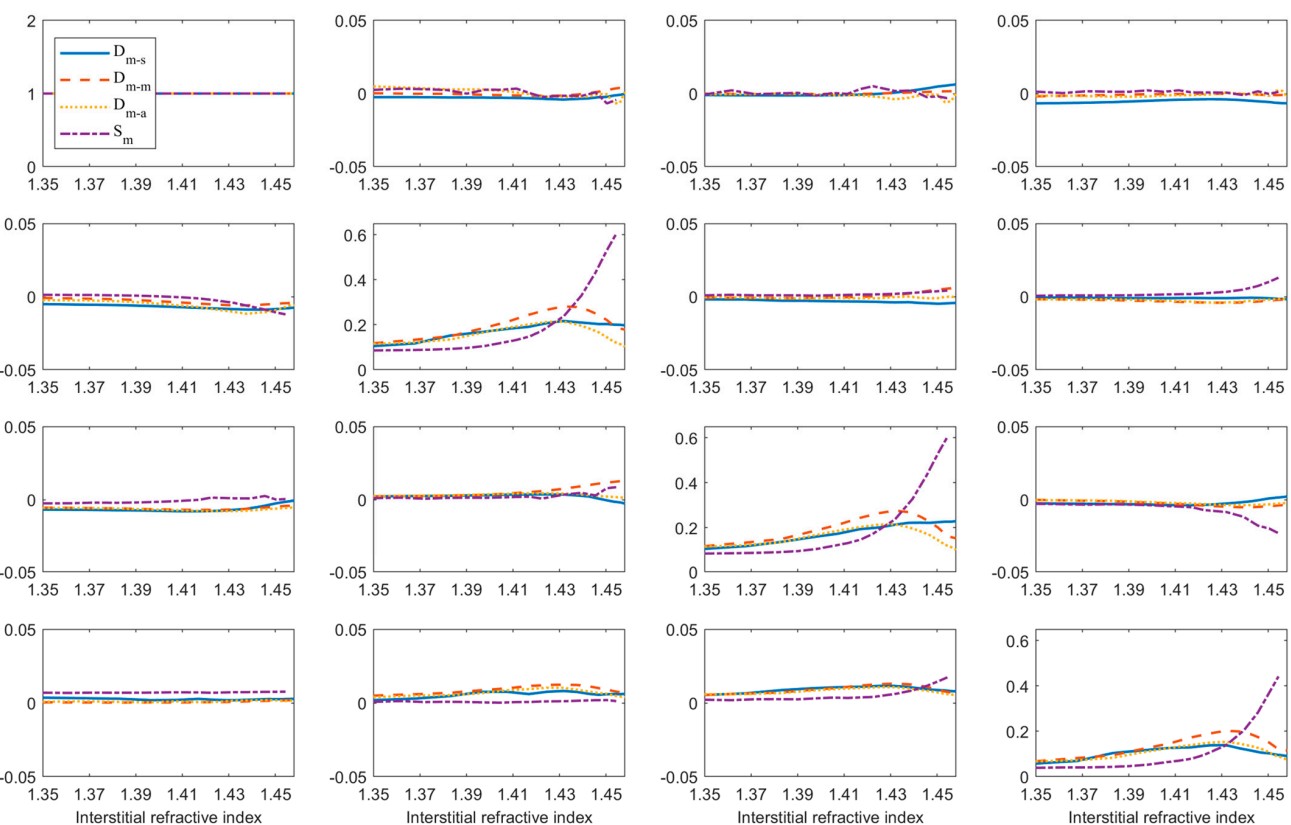

**Figure 2.** Variation of Mueller matrix elements of different samples with the TOC process. Subfigures represent $4 \times 4$ Muller matrix elements at corresponding locations, i.e., the subfigure in *i*-th row and *j*-th column represent the $m_{ij}$ element of the Mueller matrix. Each data point represents the average of the original data. The horizontal axis represents the interstitial refractive index, and the vertical axis represents the values of each element.

For the diagonal elements, the curve of the single layer milk samples $S_m$ always maintained an increasing trend, especially after the interstitial refractive index reached 1.42, wherein the slope of the curve increased significantly. This was an artifact. For media of infinitely large area and depth, the diagonal elements did not vary with changes in the scattering coefficient. The observed variations were caused by the limited volume of the sample; as the scattering coefficient decreased, an increasing number of scattered photons, particularly those scattered multiple times, were unable to be captured by the detector, which led to a reduction in depolarization and increase in diagonal elements.

The curves of double-layer samples were also maintained steadily at the beginning. The diagonal elements of double layer milk–muscle $D_{m-m}$ and milk–adipose $D_{m-a}$ samples declined in the later stage of TOC because photons penetrated into the second layer of greater depolarization capability. The lower layer of the double lay milk–silk sample $D_{m-s}$ was concentrically arranged silk encapsulated in two glass slides. It was not affected by TOC. Thus, the diagonal elements $D_{m-s}$ varied in a different trend from $D_{m-m}$ and $D_{m-a}$. As TOC progressed, the nondiagonal elements decreased, indicating an increasing contribution from the anisotropic lower layer.

### 3.1.2. Variation of PBPs during TOC

Figure 3a shows the variation of polarization parameter $b$ with the TOC for different samples. At the beginning of TOC, there were no significant differences between different samples because photons were all backscattered in the milk layer. Therefore, the depolarization caused by different samples was similar. As TOC proceeded, differences in $b$ values appeared among different samples. The $b$ value of single-layer samples increased sharply, indicating that glycerol was an effective optical clearing agent (OCA) that reduced the scattering of milk and allowed photons to retain more of their original polarization state. The lower layer of the double-layer samples were different biological tissue phantoms, and the TOC allowed photons to penetrate the milk of the upper layer, reach the lower layer, and carry information from both layers upon return. As the TOC progressed to the later stages, the scattering of milk was very small, and the polarization characteristics of the backscattering photons were almost entirely due to the lower layer. We found that when the interstitial refractive index was 1.458, the $b$ value of $D_{m-a}$ was the lowest, followed by $D_{m-m}$, and the $b$ value of $D_{m-s}$ was the highest, indicating that the depolarization caused by porcine adipose samples was the most significant, followed by porcine skeletal muscle samples, with the depolarization caused by silk samples being the weakest.

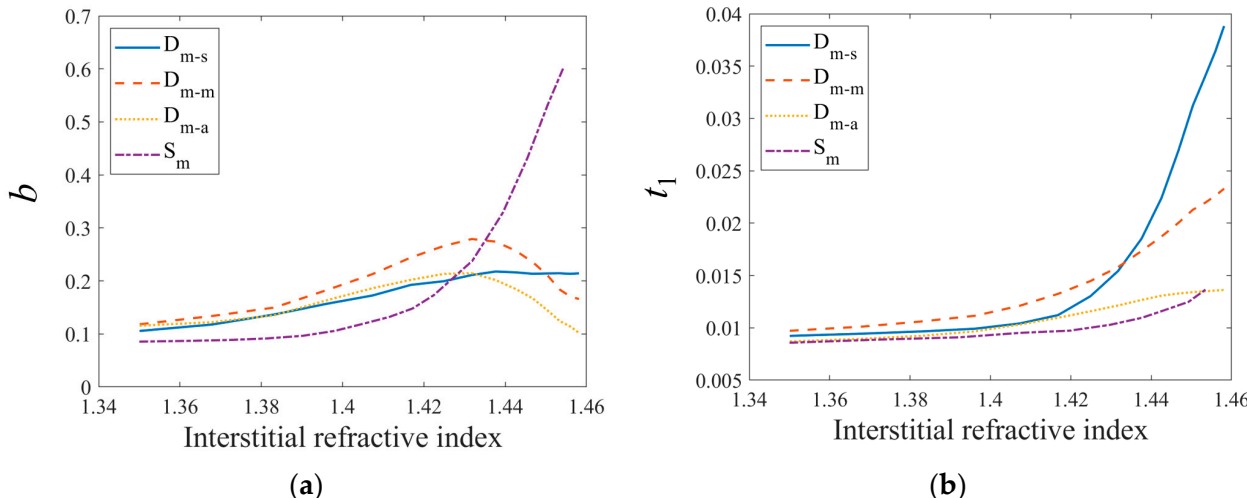

**Figure 3.** Variation of polarization basis parameters of different samples with the TOC process. Each data point represents the average of the original data. The horizontal axis represents the interstitial refractive index. (**a**) MMT parameter $b$; (**b**) MMT parameter $t_1$.

The results demonstrate that polarization parameter $b$ is effective in distinguishing between single-layer and double-layer samples. In the case of single-layer samples, decreases in the scattering coefficient to the TOC led to a gradual increase in $b$. For double-layer samples, backscattered photons may come from different layers, reflecting distinct structures in the upper and lower layer of the sample. Consequently, $b$ was affected not only by the TOC but also by the different contributions from the two layers. Since the lower layers of $D_{m-m}$ and $D_{m-s}$ were anisotropic, differing significantly from the isotropic milk in the upper layer, the temporal curves of $b$ varied sharply during the TOC as photons start penetrating into the lower layer. We recognize that $b$ can serve as a reference for distinguishing between single-layer and double-layer samples, being particularly effective for samples with substantial differences in the scattering coefficient between the upper and lower layers.

The parameter $t_1$ reflects the anisotropic properties of the sample [28]. The $t_1$ value of an isotropic sample is close to zero. Figure 3b shows the $t_1$ values of different samples as a function of the interstitial refractive index. Both porcine adipose and milk are isotropic samples, which means their polarization properties are independent of the sample's orientation. Therefore, the curves of $D_{m-a}$ and S always maintained lower values, and the curves of other samples also maintained lower values in the early stage of the TOC. The porcine skeletal muscle sample contained aligned muscle fibers, so it had anisotropy. The anisotropy of the well-aligned silk fibers was even more pronounced. As the TOC decreased the scattering of milk, the anisotropy in the lower layer of $D_{m-m}$ and $D_{m-s}$ gradually showed up in the curves. As the interstitial refractive index increased, the $t_1$ values of $D_{m-m}$ and $D_{m-s}$ started to reflect the influence by the anisotropic lower layers. It can be seen that the anisotropy of silk was much higher than that of porcine skeletal muscle. It is noteworthy that the sample exhibited weak anisotropy, even in the early stages of the TOC. This phenomenon arose from the oblique incident illumination, which induces anisotropy in an isotropic system. For the single-layer isotropic $S_m$ sample, $t_1$ kept increasing as the TOC proceeded because the increases in the penetration depth due to the TOC further enhanced the anisotropic spatial distribution in the backscattered photons in the oblique incidence configuration. For the double-layer milk–silk sample $D_{m-s}$, the oblique incident illumination led to different effects at different regions of the sample. The influences of oblique incident illumination are discussed in detail in Section 3.2.2.

The curves of anisotropy parameter $t_1$ during the TOC contained the characteristic double-layer features for samples, with an isotropic upper layer and anisotropic lower layer. Observing the temporal curves of $t_1$ for $D_{m-m}$ and $D_{m-s}$ samples allowed us to determine when photons reached the lower layer

In this section, we discussed the results of the polarization parameter $b$ and $t_1$, finding that the polarization parameters can record changes in the microstructure of the sample during the TOC process, encompassing information about the layered structure of the sample. In Section 3.2, we delve deeper into extracting structural information from the polarization parameters.

### 3.2. Extracting Microstructural Information of the Upper and Lower Layers via the TOC Process

3.2.1. Changes in PBP Images during TOC

The concentrically aligned silk was anisotropic. Its azimuthal orientation varied continuously, and its degree of alignment also varied slightly in the radial direction. Figure 4 shows images of different PBPs during TOC. Each row with the same color bar shows changes in the polarization parameter with the increase in the interstitial refractive index. When the TOC started, PBP images were all uniform, representing the properties of the top layer milk. As the TOC proceeded, effects due to the lower layer gradually emerged, and the image of $m_{22} - m_{33}$ (Figure 4b) gradually became an eight-petal pattern, with the pixel values in different directions alternating positively and negatively, corresponding to changes in the azimuth of the silk fibers [29]. The anisotropy parameter $t_1$ image (Figure 4c) exhibited a ring-shaped pattern, indicating concentrically aligned silk fibers within the

imaging area. However, it is evident that the values exhibited distinct variations between vertical and horizontal directions. The non-uniform spatial distribution in the images of $t_1$ can be explained by the additional anisotropy due to oblique incidence illumination, which enhanced or cancelled out $t_1$ of silk fibers aligned in the two orthogonal directions. We discuss the phenomenon in detail in Section 3.2.2.

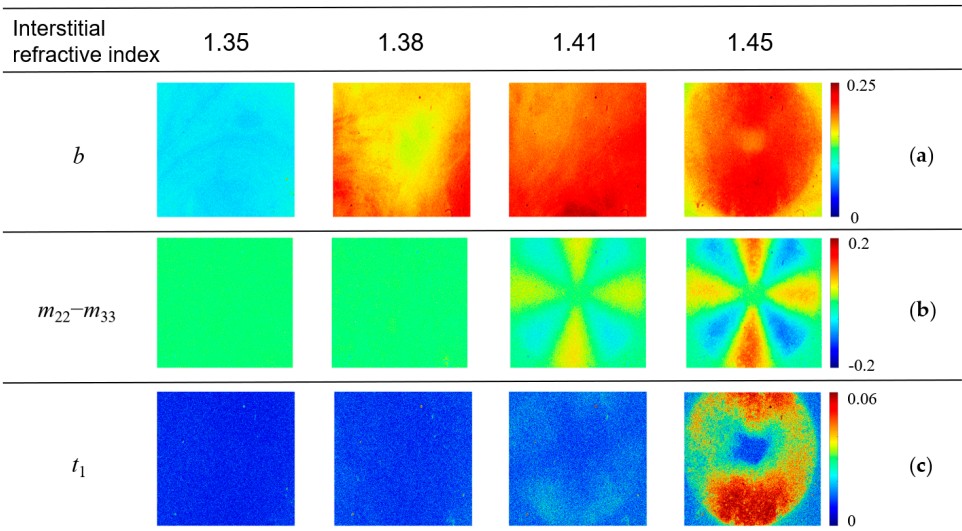

**Figure 4.** Images of polarization basis parameters (**a**) $b$, (**b**) $m_{22}-m_{33}$, and (**c**) $t_1$ of sample $D_{m\text{-}s}$ at different interstitial refractive indices.

### 3.2.2. Influence of Oblique Incident Illumination

In previous studies, we found that oblique incidence illumination can cause artifacts in polarization parameters [30]. To examine such artifacts, we carried out a regional analysis of PBP figures. By taking advantage of the different effects of oblique incident illumination on different regions, we were able to evaluate the magnitude of the variation on the polarization parameter caused by oblique incident illumination and isolate it.

There were differences in the polarization properties of different regions of the annular silk fibers. We selected 12 ROIs where silk fibers were aligned in similar directions, took the average of the polarization parameter values within each ROI, then analyzed their changes with the interstitial refractive index (Figure 5a). We first divided the annular silk fibers into four parts in azimuth directions: top, bottom, left, and right, denoted as $T$, $B$, $L$, $R$, respectively, and then divided each part radially into three regions labeled with a subscript $i$ ($i$ = 1, 2, 3) following the inward direction.

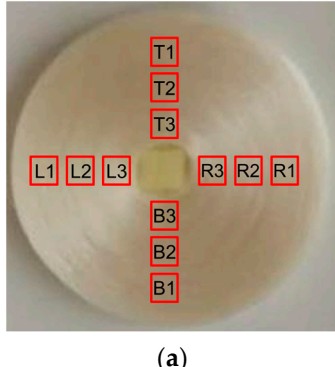

(**a**)

**Figure 5.** *Cont.*

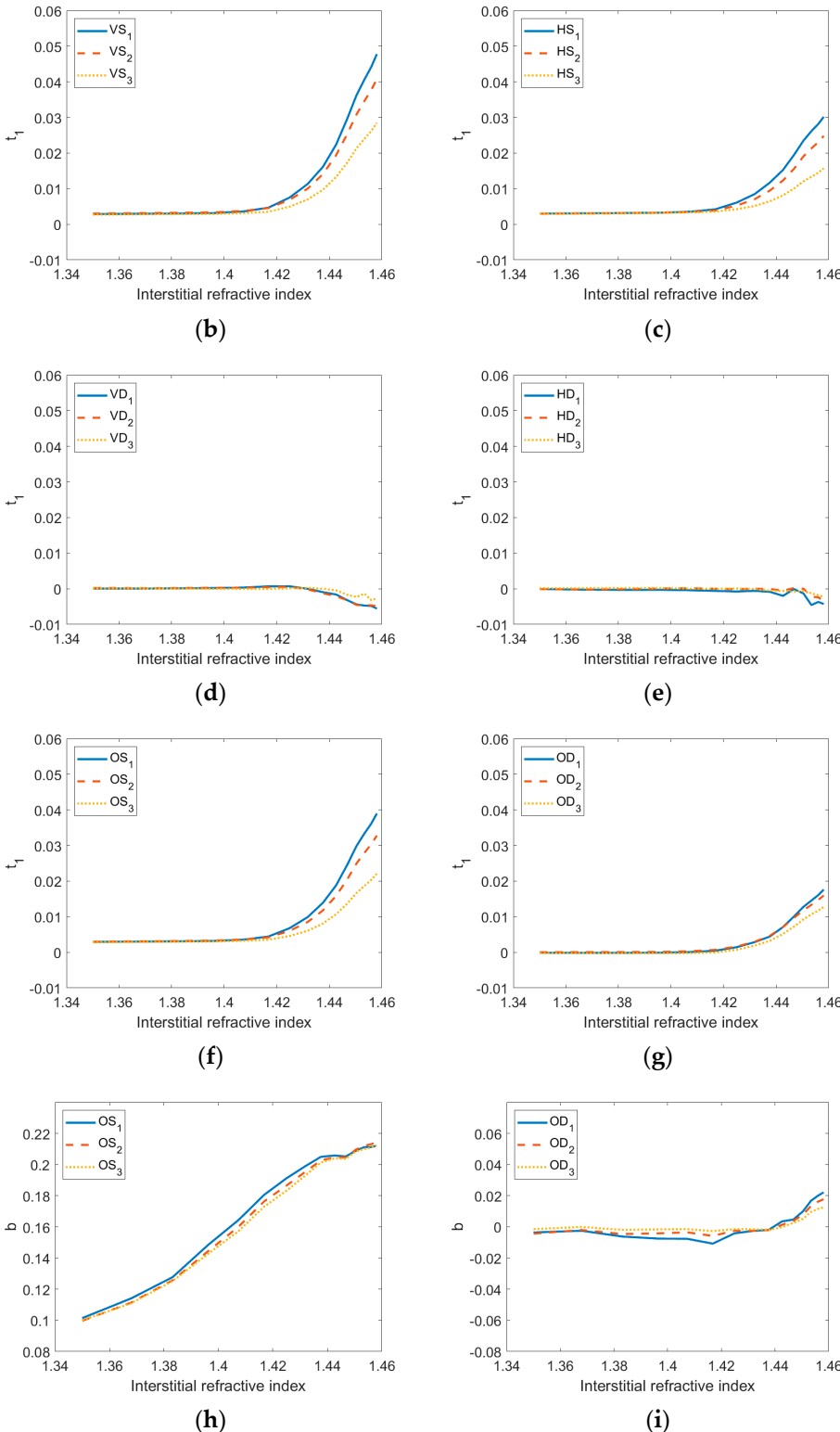

**Figure 5.** Regional analysis of PBPs. (**a**) A picture of concentrically aligned silk fibers. (**b**–**g**) The VS, HS, VD, HD, OS, and OD results of parameter $t_1$, respectively. (**h,i**) The OS and OD results of parameter *b*. The horizontal axis represents the interstitial refractive index.

We analyzed the differences in polarization parameters between the vertical (regions *T* and *B*) and horizontal (regions *L* and *R*) directions using summation and difference methods and then performed an overall analysis of the regions in both directions, as shown in Table 2. As parameter *b* and $t_1$ had better performance in differentiating upper- and

lower-layer of the samples, we analyzed their results among different regions, as shown in Figure 5b–i.

**Table 2.** Regional analysis results.

| Regions of Analysis | Definition |
|---|---|
| Vertical analysis | Summation: $VS_i = (T_i + B_i)/2$ <br> Difference: $VD_i = T_i - B_i$ |
| Horizontal analysis | Summation: $HS_i = (L_i + R_i)/2$ <br> Difference: $HD_i = L_i - R_i$ |
| Overall analysis | Summation: $OS_i = (HS_i + VS_i)/2$ <br> Difference: $OD_i = HS_i - VS_i$ |

The curves of the polarization parameter $t_1$ in different regions were plotted, with all graphs using the same coordinate axis range. The VS (Figure 5b) and HS (Figure 5c) curves were almost zero when TOC just began, and then they increased rapidly after the inflection point, with VS being overall higher than HS. This was because the direction of anisotropy due to oblique illumination was parallel to the anisotropy direction caused by the horizontally aligned silk fibers in the vertical regions, causing an increase in total anisotropy. In the horizontal regions where the silk was aligned vertically, directions of anisotropy due to the oblique incident light and fibers were perpendicular, causing the total anisotropy to decrease.

Overall analysis can evaluate the influence of oblique illumination. The OS curve (Figure 5f) represents the sample's own anisotropy, and the OD curve (Figure 5g) represents the influence of oblique illumination on the polarization parameter $t_1$. There were significant differences between the $OS_i$ curves ($i$ = 1, 2, 3). The anisotropy after the inflection point was strongest for $OS_1$, followed by $OS_2$ and $OS_3$, corresponding to the decreasing radii. In regions with larger radii, silk fibers were better aligned with smaller variations in their directions, so the anisotropy was stronger. In contrast, regions with smaller radii corresponded to better alignment of the silk fibers and weaker anisotropy. The differences between the $OD_i$ curves were small, indicating that the anisotropy caused by oblique illumination was close to uniform in different regions for the almost parallel illumination beam.

The uniformity of illumination can also be measured by taking the difference between regions with the same radius in the horizontal/vertical direction to obtain the VD (Figure 5d) and HD (Figure 5e) curves. The anisotropy on the right was slightly larger than that on the left, and the anisotropy on the bottom was slightly larger than that on the top, but the difference was very small.

Unlike $t_1$, the sensitivity of $b$ mainly came from the changes in the upper layer of the sample. The OS curve of $b$ (Figure 5h) rose rapidly at first and then became stable. This indicates that in the later stage of the TOC, the backscattering photons mainly carried information from the lower layer, and the lower layer was not affected by the TOC. The OD curve (Figure 5i) showed that oblique illumination had almost no effect on depolarization.

There were two main reasons why polarization parameters were sensitive to double-layer structures during the TOC process. One was that the parameters were sensitive to changes in the upper layer, and the differentiation between the upper and lower layers depended on the changes in the milk caused by the TOC process. The other reason was that the parameters were sensitive to the different contributions from the upper and lower layers as photons penetrated deeper into the lower layer.

Therefore, for the double-layer phantom with an isotropic upper layer and anisotropic lower layers, both $b$ and $t_1$ were able to be utilized for identifying the layered structure of the samples. The variations in $b$ and $t_1$ reflected changes in the polarization information carried by photons as they penetrated from the upper layer to the lower layer during the TOC process.

### 3.3. Probing Layered Structures in Living Skin Samples

What we learned from the experiments on double-layer phantoms was used for exploring the structure of layered tissues such as living skin.

A total of 200 sets of MM data were measured for 50 min right after the glycerol was applied on the living skin. All MM elements were normalized by $m_{11}$. By averaging the pixels in the region of interest (ROI) of each MM image, the temporal variation of each MM element was obtained. The dynamic curves of each MM element are plotted in Figure 6.

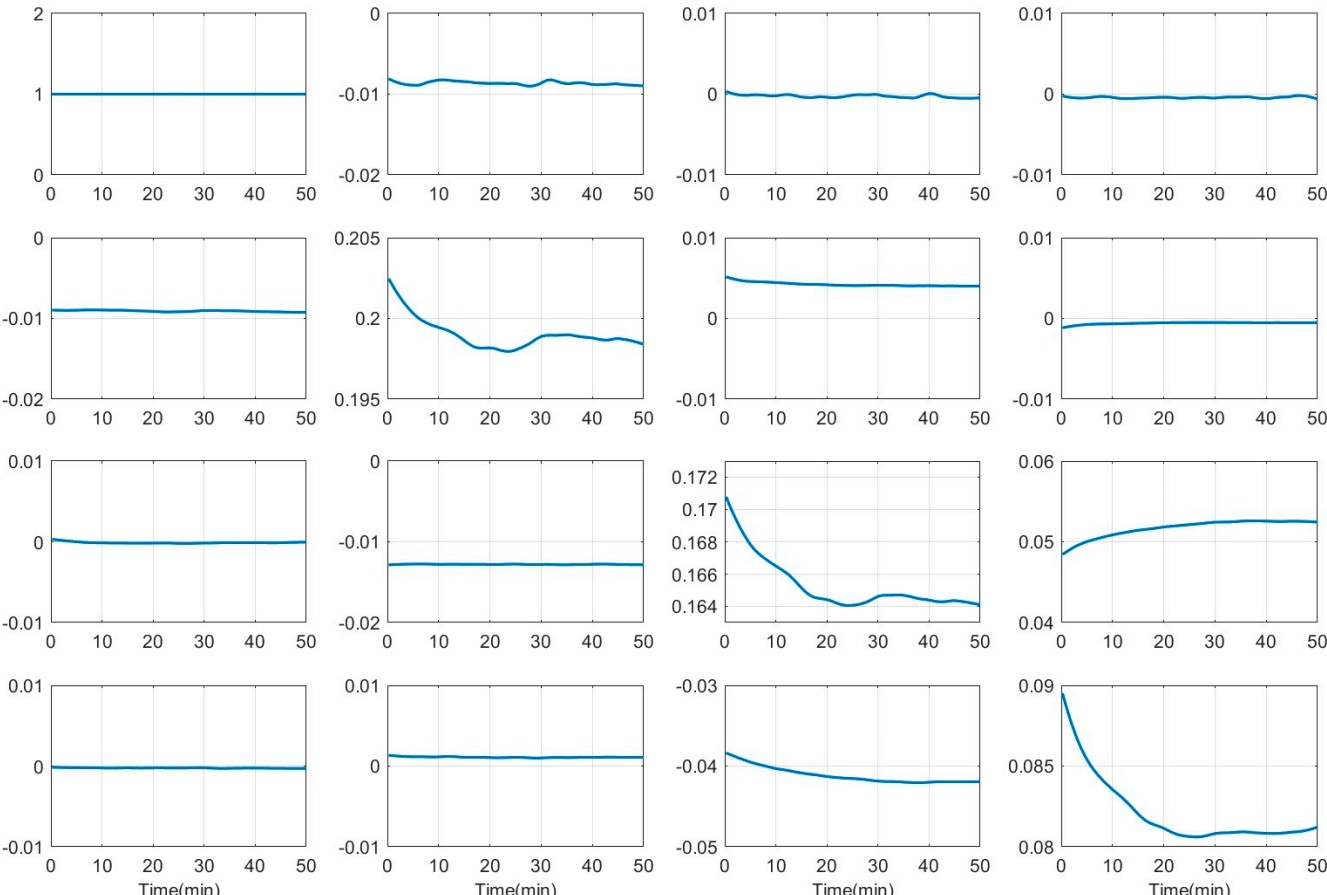

**Figure 6.** Changes in the elements of the Mueller matrix (MM) within 50 min after the application of glycerol. Subfigures represent $4 \times 4$ Muller matrix elements at corresponding locations. The horizontal axis represents time (in minutes), and the vertical axis represents the values of the matrix elements.

From Figure 6, it can be observed that the changes in the Mueller matrix (MM) during the TOC mainly manifested in the diagonal elements ($m_{22}$, $m_{33}$, $m_{44}$). The diagonal elements initially decreased at the beginning of the TOC and then tended to stabilize. At around 20 min, the curve exhibited a turning point, which indicates a multilayer structure in the mouse skin [22]. The elements $m_{34}$ and $m_{43}$ also showed changes, with their absolute values gradually increasing during the TOC. This suggests the presence of some anisotropic property of the skin. The differences between $m_{22}$ and $m_{33}$ are related to the degree of alignment, or anisotropy, of the fibrous structure arrangement in the skin [29].

To further explain the relationships between the polarization and microstructural features, we analyzed the temporal behaviors of PBPs of the mouse skin during the TOC. By averaging PBP values of pixels in the ROI, the temporal curves of parameters $b$ and $t_1$ were plotted, as shown in Figure 7.

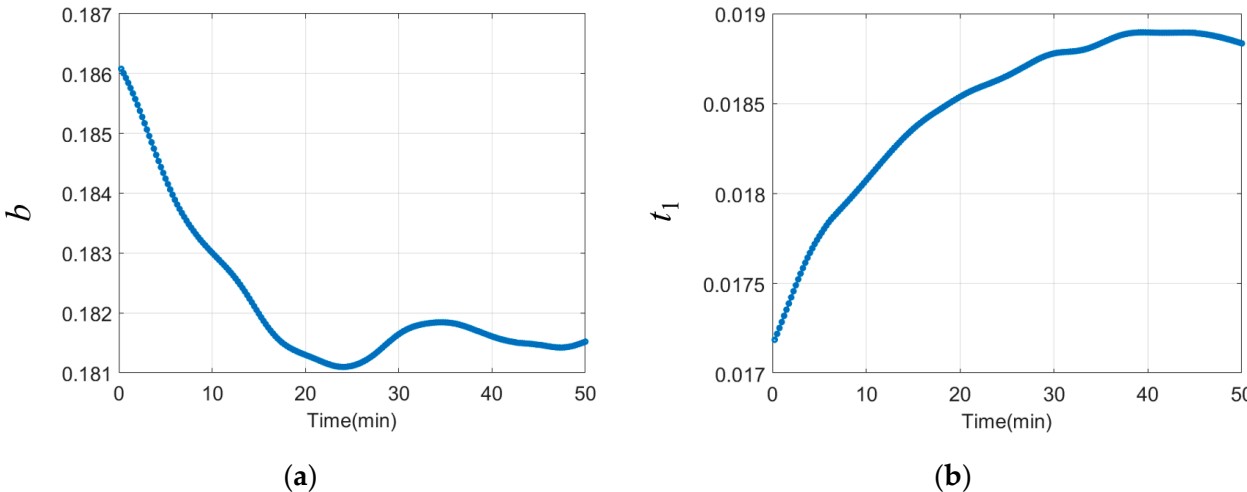

**Figure 7.** Changes in the PBPs within 50 min after the application of glycerol. The horizontal axis represents time (in minutes), and the vertical axis represents the values of PBPs. (**a**) MMT parameter $b$; (**b**) MMT parameter $t_1$.

Figure 7a shows that $b$ gradually decreased after the TOC and reached the minimum at 25 min. This suggests a layered structure in the mouse skin with different depolarization properties. As photons penetrated deeper in the top layer, the scattering depolarization increased. The turning point at 25 min indicates that the depolarization capability of the bottom layer should be lower than that of the top layer. Photons penetrating into the bottom layer led to a deviation from the single-layer behavior in the curve of $b$ during the TOC.

Figure 7b shows that $t_1$ continued to increase after 20 min of the TOC, with a slower rate of increase after 20 min, and it tended to stabilize after 36 min. The temporal behavior of parameter $t_1$ confirmed the presence of a layered structure and revealed the anisotropy of the lower layer skin.

The results of phantom experiments indicate that the polarization parameters $b$ and $t_1$ were effective indicators sensitive to the complex double-layer structures. The findings from living skin experiments suggest a similar double-layer structure in the skin, with the upper layer being isotropic and the lower layer being anisotropic. Unlike phantom experiments, the TOC had a more pronounced effect on living skin, as $t_1$ revealed anisotropic information of the lower layer within the first few minutes of the TOC, and the time curve of $b$ deviated from the behavior of single-layer samples right from the beginning. The living skin experiment once again highlights the ability of polarization parameters to characterize the dynamic TOC process in tissues and extract information about layered structures.

### 4. Conclusions

This study observed the temporal variations in Mueller matrix imaging data of phantoms with TOC. Different phantoms exhibited different dynamic behaviors in polarization data during the TOC, which were correlated to different structure features of the layers. Polarization parameter $b$ was able to effectively distinguish single-layer or double-layer samples with scatterers of different sizes, while $t_1$ was more sensitive to double layers of different anisotropies. Using physically meaningful PBPs and their images for more detailed examination the $D_{m\text{-}s}$ sample, we observed the differences between regions in PBP images due to oblique incident illumination. We also conducted a regional analysis to further investigate the impact of oblique incident illumination on measurement results and explained the mechanism behind the sensitivity of $b$ and $t_1$ to the double-layer structure. The results indicated that $b$ was sensitive to changes in the upper layer (isotropic layer) and was not influenced by oblique incident illumination. On the other hand, $t_1$ reflected the

anisotropic structure of the lower layer and was affected by oblique incident illumination but can be compensated for.

We conducted TOC experiments on living skin. We discovered a double-layer structure in living skin, and with the progression of the TOC, the anisotropic structural information of the lower layer gradually became apparent.

The polarization parameters reflected the microstructure of the sample. By observing the changes in polarization parameters during the TOC process, the layered structural features of the sample were able to be characterized. Our work demonstrates the promising application of the combination of backscattering polarimetry and TOC in biological tissues, paving the way for the real-time, non-invasive, and in situ detection of living skin diseases.

**Author Contributions:** Conceptualization, H.M.; data curation, Q.L., T.B., Y.S., and Y.W.; formal analysis, Q.L.; methodology, T.B. and T.H.; resources, Y.S. and Y.W.; supervision, Y.W. and H.M.; writing—original draft, Q.L.; writing—review and editing, Q.L., T.B., and H.M. All authors have read and agreed to the published version of the manuscript.

**Funding:** This research was funded by National Natural Science Foundation of China, grant numbers 11974206 and 62375152.

**Institutional Review Board Statement:** The animal study protocol was approved by the Ethics Committee of the China Academy of Chinese Medicine Science (protocol code: ERCCACMS21-2111-27).

**Data Availability Statement:** The data presented in this study are available on request from the corresponding author. The data are not publicly available due to privacy.

**Conflicts of Interest:** The authors declare no conflicts of interest.

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
