# Peer review of "Probing Layered Tissues by Backscattering Mueller Matrix Imaging and Tissue Optical Clearing"

_photonics, doi:10.3390/photonics11030237_

Round 1

Reviewer 1 Report

Comments and Suggestions for Authors

In the manuscript "Probing layered tissues by backscattering Mueller matrix imaging and tissue optical clearing" by Qizhi Lai et al.,tissue optical clearing is used to increase optical penetration depth and characterize the layered structures of tissue samples. Different tissue phantoms were constructed to examine changes in the polarization features of the layered structure during tissue optical clearing process. It is found that depolarization and anisotropy parameters can distinguish between single-layer and double-layer phantoms, reflecting microstructural information from each layer. Finally, living skin samples have been used for experiments. As authors concluded, the results show that the combination
of backscattering polarization imaging and tissue optical clearing provides a powerful tool for the characterization of layered samples.
The paper is well and clearly written with good English. The title and abstract are adequate and the introduction is informative. The obtained results are interesting and useful. I recommend the publication of the article after correction of a couple of misprints.

l 109: The lower layers of are ?

l 245: the lowers layer. --> the lower layer.

l 348: the lower layer s is not --> the lower layer is not

Comments on the Quality of English Language

The English is good. Some minor corrections are indicated

Author Response

Thank you very much for your valuable comments, I have gained a lot from your comments.

The attached document is a response to your comments.

Reviewer 2 Report

Comments and Suggestions for Authors

Major concerns:

1.        What is the previous work in Line 51?

2.        One huge issue for this article is the presentation of the figures. The authors should go through the article to make sure that the y-axis tittle is provided.

3.        Figure 2 and Figure 6 are very confusing. What are the conditions for the subfigures? What do the subfigures represent? This information should be provided in the caption.

4.        Figure 4 lacks the legend.

Comments on the Quality of English Language

1.        When demonstrating the experiment procedures, both present tense and past tense were used. This causes difficulties in reading and understanding.

2.        Line 180 is the illustration from the template, which should be deleted.

Author Response

(The authors gave the same response as above.)

Reviewer 3 Report

Comments and Suggestions for Authors

The authors have demonstrated a unique technique to probe the layered structural information of tissues (both phantoms and living cells) by using backscattering Mueller matrix imaging. They have used tissue optical clearing (TOC) technique to increase optical penetration depth and characterize the layered structures of tissue samples, unlike using a multiwavelength approach. All in all, I am really convinced by the technical details, results and presentation of their studies and recommend to be published in Photonics. However, I do have few questions that the authors can perhaps consider discussing.

I believe one important aspect of TOC technique is to use OCA (optical clearing agents) for refractive index matching and to enhance the imaging (depth related) capabilities of this technique. However, how practical is this technique in real life – perhaps live imaging of biological tissues, say for humans? Use of OCA dictates use of an external agents to capture the images, but this can introduce complexity of any allergic reaction to glycerol or any other agent used as OCA? So may be not so non-invasive at the end ?

There is already an established technique of OCT (Optical Coherence Tomography), which I believe has found many real world practical applications in imaging layered structure and also the structural depth of information? How does this presented technique of back scattering Mueller matrix imaging based on TOC fair/compare to OCT?

Can the authors comment on possibility of using a white light source/supercontinuum and apply their current technique to further enhance the sensitivity of their presented technique ?

Author Response

(The authors gave the same response as above.)
